# In Vitro Growth, Receptor Usage and Pathogenesis of Feline Morbillivirus in the Natural Host

**DOI:** 10.3390/v14071503

**Published:** 2022-07-08

**Authors:** Veljko Nikolin, Leticia Hatsue Sobreda Doi, Michael Sieg, Johannes Busch, Denny Böttcher, Laurence Tedeschi, Amélie Poulard, Vincent Staszewski, Thomas Vahlenkamp, Herve Poulet

**Affiliations:** 1Boehringer-Ingelheim Vetmedica, Binger Str. 173, 55218 Ingelheim am Rhein, Germany; veljko.nikolin@boehringer-ingelheim.com (V.N.); lehatsue1@gmail.com (L.H.S.D.); 2Institute of Virology, Center for Infectious Diseases, Faculty of Veterinary Medicine, Leipzig University, An den Tierkliniken 29, 04103 Leipzig, Germany; michael.sieg@vetmed.uni-leipzig.de (M.S.); johannes.busch@vetmed.uni-leipzig.de (J.B.); vahlenkamp@vetmed.uni-leipzig.de (T.V.); 3Institute of Veterinary Pathology, Faculty of Veterinary Medicine, Leipzig University, an den Tierkliniken 33, 04103 Leipzig, Germany; denny.boettcher@vetmed.uni-leipzig.de; 4Boehringer-Ingelheim, Cours du 3ème Millénaire, 69800 Saint-Priest, France; laurence.tedeschi@boehringer-ingelheim.com (L.T.); amelie.poulard@boehringer-ingelheim.com (A.P.); vincent.staszewski@boehringer-ingelheim.com (V.S.)

**Keywords:** feline morbillivirus, CD150, SLAMF1, cat, kidney, tubulointerstitial nephritis

## Abstract

Feline morbillivirus (FeMV) is a recently discovered virus belonging to the genus *Morbillivirus* of the virus family *Paramyxoviridae*. Often, the virus has been detected in urine of cats with a history of urinary disease and has a worldwide distribution. Currently, it is unclear which receptor the virus uses to enter the target cells. Furthermore, many aspects of FeMV biology in vivo, including tissue tropism, pathogenesis, and virus excretion in the natural host remain unclear. In this study we analyzed the replication of FeMV in various cell lines. Secondly, we tested if the presence of feline SLAMF1 (Signaling Lymphocytic Activation Molecule family 1/CD150, principal entry receptor for other members of the *Morbillivirus* genus) improved FeMV replication efficiency in vitro. Finally, to elucidate in vivo biology in cats, as a natural host for FeMV, we experimentally infected a group of cats and monitored clinical symptoms, viremia, and excretion of the virus during the course of 56 days. Our study showed that FeMV shares some features with other morbilliviruses like the use of the SLAMF1 receptor. For the first time, experimental infection of SPF cats showed that FeMV does not induce an acute clinical disease like other morbilliviruses but can induce lesions in the kidneys, including tubulointerstitial nephritis. Further investigations are needed to confirm the site and dynamics of replication of FeMV in the urinary tract and the longer-term impact of FeMV-induced lesions on the renal function. Whether FeMV infection can result in chronic kidney disease will require the monitoring of cats over a longer period.

## 1. Introduction

FeMV was first reported in 2012 in stray cats from Hong Kong [1]. Since then, FeMV has been detected within cat populations in Japan [2,3], Germany [4], Italy [5], United States [6], Brazil [7], Turkey [8], the United Kingdom [9], Malaysia [10], and mainland China [11]. Wide geographical distribution suggests a global presence of the virus. Depending on the cat population, FeMV RNA was reported in a range of 3–53% of the analyzed urine samples, suggesting a high overall prevalence of the virus in cat populations.

FeMV has been suspected to be associated with tubulointerstitial nephritis (TIN), the histopathologic correlate of idiopathic chronic kidney disease (CKD) [1,2,4,5,12,13]. Indeed, in feline CKD, typical histologic features include interstitial inflammation, tubular atrophy, and fibrosis with secondary glomerulosclerosis. In contrast to human and dog CKD, tubulointerstitial fibrosis is a hallmark of feline CKD pathogenesis in early and advanced stages of disease, whereas primary glomerulopathy with marked proteinuria is a rare finding in cats [14,15,16]. CKD is the most common metabolic disease of domesticated cats, affecting 30–40% of cats older than 10 years and is an important cause of morbidity and mortality in cats [17]. Although several factors have been correlated to the progression of CKD in cats, including hyperphosphatemia, proteinuria, anemia, systemic hypertension, aging, and tissue hypoxia, there is a lack of direct evidence to suggest that these factors play a role as an initial cause of CKD [18]. In most cases, no primary cause has been identified and finding risk factors may help with early detection, understanding the disease pathogenesis, and development of new treatments.

The link between FeMV infection and CKD has not been proven yet, but suggested by several observations: (i) a prevalence of 6.1% in urine and 40% (4/10) in kidney tissues of cats with nephritis [2], (ii) a prevalence of 6.7% (8/120) in cats with signs of urinary tract disease in contrast to no viral detection in healthy cats (*n* = 86) [4], (iii) a high correlation (90%, 26/29) between inflammatory renal lesions and FeMV infection in comparison to non-infected cats (62%, 44/71) [12], and (iv) a significant association between the presence of FeMV antigen and morphologic tubular and interstitial alterations in cat kidneys [13]. Recently, histopathological examination of two cats naturally infected with FeMV showed diffuse renal tubular vacuolation with multifocal membranous glomerulo-nephropathy and multifocal necrotizing hemorrhagic cystitis [19]. Furthermore, the same study showed histological evidence of FeMV matrix protein presence and pathologic changes in lungs, brain, liver, and gastro-intestinal organs. By RT-qPCR, high RNA loads were measured in urine samples, moderate RNA loads in spleen, kidney, lungs, urinary bladder, and intestines, while no FeMV could be detected in liver and brain. These findings indicate that the virus may have a systemic distribution; however, the long-term persistence, viral excretion, and pathogenesis are mainly associated with urinary tract infection. In contrast, other studies could not find any evident association between FeMV infection and histopathological alterations or renal failure, as the virus could be detected in cats without clinical symptoms of urinary disease [7,8,9]. Therefore, a possible causal role of FeMV infection in pathogenesis of CKD in cats remains unclear.

Morbilliviruses are usually transmitted via the respiratory route, infecting the host through inhalation of respiratory droplets or aerosols by direct contact with body fluids or fomites. After inhalation, virions infect target cells by binding of the H glycoprotein to cellular receptors and then enter the cell by fusing their envelope with the cell membrane. Two cellular receptors have been described for wild-type morbilliviruses: (i) signaling lymphocyte activation molecule family member 1 (SLAMF1) and (ii) nectin cell adhesion molecule 4 (nectin-4, previously known as poliovirus receptor-like 4). SLAMF1 represents a set of universal receptors for morbilliviruses [20,21]. This suggests that FeMV may use the same receptors for entry into susceptible cells. Furthermore, Sakaguchi et al. [22] reported that FeMV infected diverse feline cell lines including epithelial, fibroblastic, lymphoid, and glial cells, indicating that its receptor or receptors are ubiquitously present on a variety of cat cells. SLAMF1 and nectin-4 are, however, not simultaneously present on mammalian cells, as SLAMF1 is expressed on subsets of immune cells and nectin-4 is found in the junctional complex of epithelial cells. The localization of those receptors explains the tropism of morbilliviruses for immunological and epithelial cells, and the associated clinical diseases [21,23,24,25]. There is currently no information available regarding the type of receptors involved in FeMV infection.

SLAMF1 was discovered in 1995 [26] on peripheral-blood T cells, T-cell clones, immature thymocytes, and a proportion of B cells. Tatsuo et al. [20] first reported SLAMF1 as a cellular receptor for Measles virus (MeV) entry in non-susceptible cells transfected with a human SLAM cDNA. Subsequent studies demonstrated that SLAMF1 is a universal receptor for morbilliviruses [21,23,24]. SLAMF1 is expressed on thymocytes, activated T- and B-lymphocytes, mature dendritic cells (DCs), macrophages, and platelets and is also a marker for the most primitive hematopoietic stem cells [27,28]. It has been implicated in lymphocyte development, thymocyte maturation, and immunological synapse formation, having a broad involvement in the modulation of innate and acquired immune responses [28].

SLAMF1 exhibits a relatively low level of sequence conservation between species. The feline SLAMF1 receptor shows a 66% sequence identity at the amino acid level to the human SLAMF1 receptor [29]. Among carnivores, felid SLAMF1 sequences (African lion and domestic cat) share a high (96.3%) amino acid sequence identity. They represent a cluster phylogenetically distant to canid SLAM sequences, having a low similarity (73.7–74.9%) with SLAMF1 from canid species (domestic dog, raccoon dog, and red fox). Despite those differences, morbilliviruses are able, with a various efficiency, to bind and initiate membrane fusion using a wide variety of SLAMF1 molecules [29].

Two genotypes of FeMV have been identified so far, genotype 1 including two clades and genotype 2 [30]. FeMV genotype 2 (FeMV-GT2) Gordon strain was isolated from a cat, Gordon, with CKD (IRIS stage 2) [31]. Using this strain, our objective was to study the characteristics of FeMV growth in cell culture, its affinity to use the SLAMF1 receptor and importantly, to decipher some aspects of virus biology in cats after experimental infection.

## 2. Materials and Methods

### 2.1. Cell Culture

CRFK (ATCC CCL-94) cells were cultured in minimal essential medium with Earle’s modified salts (MEM-E) which was supplemented with 10% (*v*/*v*) heat-inactivated fetal bovine serum (FBS). LLC-MK2 cells (ATCC CCL-7), Vero cells (ATCC-CCL81), Vero cells stably expressing canine SLAMF1 (VeroDogSLAM), and Vero cells stably expressing feline SLAMF1 (VeroCatSLAM) [29] were cultured in MEM-E which was supplemented with 5% heat-inactivated FBS. Cells were passaged twice a week. Cell seeding densities for every passage were 1–2 × 10^4^ cells/cm^2^. VeroCatSLAM and VeroDogSLAM cells were treated with selection antibiotic Zeocin (Thermo Fisher Scientific, Waltham, MA, USA; cat. #R25001) at 0.5 mg/mL to ensure strict selection of SLAMF1-positive cells. All cells were cultivated at 37 °C in a humidified atmosphere containing 5% CO_2_.

### 2.2. Immunofluorescence Staining of Infected Cells

Infected and control cells were fixed with an 85% acetone solution at −20 °C for 30 min. Cell monolayers were dried under the airstream of the biosafety cabinet and stored at −20 °C. After one PBS wash, cell plates were incubated with 200 µL of 5% BSA in PBS per well at 37 °C for 1 h. After rinsing once with PBS, 100 µL of a 1:200 dilution of a specific anti-FeMV nucleoprotein (N) rabbit serum (Faculty of Veterinary Medicine, Leipzig University) were added to the wells (rabbit antiserum was produced by immunization with recombinant FeMV N protein produced in *Escherichia coli*). Cells were incubated for 60 min at 37 °C and then rinsed three times with PBS. A goat anti-rabbit IgG conjugated with AlexaFluor^®^ 488 secondary antibody (Thermo Fisher Scientific, cat. #A-11008, RRID AB_143165) (diluted at 1:500 in PBS) was added to the cells and incubated for 30 min at 37 °C. Finally, the cells were rinsed twice with PBS, overlayed with 100 μL for 96 multi-well plates or 500 µL for 24 multi-well plates of PBS and stored at 4 °C protected from light until fluorescence microscopy.

### 2.3. Quantitative PCR

RNA was extracted from samples (urine, nasal swabs, plasma) using a MagMAX CORE Nucleic Acid Purification Kit in a King Fisher Duo Prime system, according to the manufacturer’s recommendations. Extracted viral RNA was used with a TaqMan Fast Virus Master Mix (Thermo Fisher Scientific, cat. #4444434)in a 96 multi-well format with the StepOnePlus System, primers FMoV-2 fwd (5′-CACCAAATGATGATGATATAAC-3′) and FMoV-2 rev (5′-CAACCATCCTTTACTTAATCTA-3′) to set up reactions. A dual-labeled FMoV probe (Fam-GCT GCT GAG GCT GAA AAC CG-Tamra) was used for the targeting of the nucleoprotein of FeMV. The reaction mix was prepared by adding 5 µL of the template in a total volume of 20 µL, as recommended by the manufacturer. The reaction was prepared as follows: the 25 μL reaction volume contained 5 µL of purified RNA, 4 μL of nuclease-free water, 12.5 µL of 2× RT-PCR Master Mix, 1 μL of forward primer (final concentration: 400 nM), 1 μL of reverse primer (final concentration: 400 nM), 0.5 µL of FeMV probe (final concentration: 200 nM), and 1 µL of 25× ADN polymerase AmpliTaq™ Fast. As the negative control, TaqMan Fast Virus Master Mix with ultrapure DNAse/RNAse-free water was used. All samples were tested in duplicates. The following thermal profile was used: 1 cycle of reverse transcription at 45 °C for 10 min, 1 cycle of PCR initial activation step at 95 °C for 10 min followed by 40 cycles of 95 °C for 15 s, 56 °C for 45 s, and 60 °C for 45 s.

### 2.4. Virus Titration

Virus titrations were carried out in 96 multi-well cell culture plates seeded with VeroCatSLAM cells. The tenfold dilutions were performed on 48 multi-well cell culture plates by diluting 120 µL of the samples in 1080 µL of MEM-E for the initial dilution (10^−1^) and subsequently transferring 120 µL of the previous dilution to the next dilution until the final dilution (10^−8^). In the next step, diluted solutions were transferred in a volume of 100 µL/well in 8 technical replicates to previously prepared 96 multi-well cell culture plates seeded with 2 × 10^3^ VeroCatSLAM cells/well. After seven days of incubation at 37 °C in a CO2 incubator, read outs were performed by microscopy and positive wells were defined by the presence of a specific cytopathic effect (CPE) in the form of syncytia, similar to those observed for CDV or MeV on Vero SLAM cells [29]. The 50% tissue culture infective dose (TCID_50_) was calculated by the Reed and Muench’s method.

### 2.5. FeMV Replication Kinetics in Susceptible Cells

CRFK, Vero, VeroCatSLAM, and LLC-MK2 cells were seeded in 6 multi-well cell culture plates in duplicates at a density of 1 × 10^5^ cells/well (CRFK) or 4 × 10^5^ cells/well (Vero, VeroCatSLAM and LLC-MK2) and incubated at 37 °C for 24 h prior to infection. All cells were infected at an MOI of 0.1 and incubated at 37 °C for 7 days. Daily, wells of each cell line were completely scraped and resuspended in the cell culture supernatant present in each well. All samples were aliquoted and stored at −80 °C for subsequent RNA extraction for RT-qPCR and viral titration assays.

### 2.6. Comparison of RNA Loads of Cells Stably Expressing Dog and Cat SLAMF1

Vero, VeroDogSLAM, and VeroCatSLAM cells were seeded in 24 multi-well cell culture plates at a density of 2 × 10^4^ cells/well and incubated at 37 °C for 24 h prior to infection. After this period, cells were infected with FeMV at an MOI of 0.1. Cells were incubated at 37 °C for 8 days prior to supernatant sampling and acetone fixation of cells. All samples were aliquoted and stored at −80 °C until subsequent RNA extraction for RT-qPCR.

### 2.7. Animal Experiment

Fifteen specific pathogen-free cats aged between 35 and 45 weeks (Hill Grove strain, Centre Lago, France) were randomly allocated to 3 groups of 5 cats, to avoid too many and frequent individual samplings, for ethical reasons. The only difference between the 3 groups was the timing of the various samples (Table 1). Cats were housed together in BSL-2 facilities. On day (−11), cats were tested for FeMV antibodies (IFA), FeMV viremia, hematological (complete blood cell counts), and biochemical blood parameters (alanine transaminase, aspartate transaminase, alkaline phosphatase, urea, creatinine, serum amyloid A). On day 0, all cats were infected via an intravenous catheter with 10^4.6^ TCID50 of FeMV GT2 strain. Viral inoculum was administered with a catheter under anesthesia. The virus stock corresponded to the 11th passage on the LLC-MK2 cell line. The virus was stored at −70 °C. After infection, cats were followed daily for clinical signs (rectal temperature, weight loss, general condition, dehydration, vomiting, diarrhea, abdominal pain, nasal or ocular discharge, buccal ulcers, and any other clinical abnormality), and tested for FeMV viremia, viral excretion in the urine and in nasal swabs, FeMV immunofluorescence antibodies (IFA), hematological and biochemical blood parameters, and histopathological examination as illustrated in Table 1. Urine was collected by cystocentesis and blood was collected from the jugular vein. Animals were sedated only if necessary. Tissues were collected immediately after euthanasia. Hematological and biochemical analyses were performed at Biovelys Laboratory (VetAgroSup, Marcy l’Etoile, France). Antibody assays and quantitative RT-PCR on plasma and urine samples were performed at the Institute of Virology of Leipzig University (Leipzig, Germany). Histopathological examinations were carried out at Vet Diagnostics Laboratory (Lyon, France) by a board-certified histopathologist. The histopathologist was blind to the group allocation. This animal experiment and the associated procedures were reviewed and approved by the Boehringer-Ingelheim ethical committee and by the French Ministry of Higher Education, Research and Innovation on 29 March 2018 (APAFIS#12498-2017120816072814).

### 2.8. Immunohistology

Kidneys from FeMV-infected cats were collected and fixed in buffered 10% formalin solution. Fixed tissues were then embedded in paraffin and cut into sections (5 µm) and mounted onto SuperFrost^®^ (Thermo Fisher Scientific, cat #17294884) slices. Antigen retrieval of deparaffinized sections was completed as previously described using 0.05% citraconic anhydride at 95 °C for 30 min [32]. Following blocking with SuperBlock T20 (Thermo Fisher Scientific cat. #37536) at room temperature for 30 min, a diluted (1µg/mL) anti-FeMV-nucleoprotein antibody was applied overnight at 4 °C, as described earlier [33]. Thereafter, sections were washed 3 times with PBS and incubated with an anti-rabbit AlexaFluor^®^594-conjugated secondary antibody (Thermo Fisher Scientific, cat. #Z25307) for two hours at room temperature. Nuclei were counterstained using 4′,6-Diamidino-2-phenylindole dihydrochloride (DAPI, Carl Roth, Karlsruhe, Germany).

### 2.9. Immunofluorescence Antibody (IFA) Assay

Immunofluorescence assay was performed as described previously [34]. Briefly, every second row of LLC-MK2 cells in a 96-well dish was infected with FeMV-GT2—remaining rows were MOCK-infected and used as controls. After five days, cells were fixed with ice-cold, 85% acetone (*v*/*v*) at –20 °C for 10 min. Acetone was then removed, cells washed with PBS, and stored at 4 °C. Prior to staining, non-specific binding sites were blocked with 5% (*w*/*v*) BSA in PBS for 1 h at 37 °C. Cat sera were diluted 1:100 (*v*/*v*) using 1% BSA in PBS. For each dilution, 100 µL were added to the FeMV-GT2- and MOCK-infected wells and incubated for 1 h at 37 °C. After 3 washing steps, an anti-cat AlexaFluor^®^488-conjugated secondary antibody was used and incubated for 30 min at 37 °C. Nuclei were counterstained using DAPI (Carl Roth, Karlsruhe, Germany). Sera from cats persistently infected with FeMV-GT2 and serum samples from SPF cats served as positive and negative controls, respectively. Semi-quantitative analyses were completed by comparison of the observed fluorescence with that of different dilutions of the positive control serum and the negative control serum.

### 2.10. Virus Neutralizing Antibody Assay

Virus neutralization assay was performed as described previously [31]. Briefly, cell monolayers of LLC-MK2 cells were used at approximately 90% of confluence in a 96-well microtiter format. Feline serum samples were incubated at 56 °C for 30 min to inactivate complement. Two-fold serial dilutions (starting with a four-fold dilution) of each serum was then mixed with ~200 TCID_50_ of FeMV-GT2 Gordon strain in an equal volume of DMEM (50 µL) followed by an incubation period of one hour at 4 °C. This suspension was used for infection of LLC-MK2 cells for two hours at 37 °C. Samples were then replaced by DMEM supplemented with 2% FBS, sodium pyruvate, non-essential amino acids, and 100 IU/mL penicillin–streptomycin. Four days later, infected cells were visualized applying the described immunofluorescence protocol. Each experiment was performed in duplicates. Neutralizing titers were defined as the reciprocal of the serum dilution at which the number of plaques was reduced to 50% relative to virus-only controls. The technique was not available when the study was initiated. It was performed afterwards when serum was still available (not all cats).

## 3. Results

### 3.1. LLC-MK2 Cells and VERO Cells Stably Expressing SLAMF1 Receptor Allow Productive Infection In Vitro

Productive growth of FeMV-GT2 in vitro has been demonstrated on cells derived from domestic cat and Vero cells [22]. To further investigate the in vitro tropism of FeMV-GT2, we tested the capacity of CRFK, Vero, VeroCatSLAM, and LLC-MK2 cells to support the replication of the virus. We used quantitative RT-PCR to monitor kinetics of RNA loads in the wells, on a daily basis during seven days of incubation. Viral RNA loads of FeMV-infected CRFK cells did not show any increase (CT mean 20.9) on day 1 after infection and remained low until the end of experiment (Figure 1A). Despite slightly higher viral loads on Vero cells at day 1 post infection (CT mean 15.5), no increase in virus loads was detected on Vero cells in the following days. In contrast, VeroCatSLAM- and LLC-MK2-infected cells showed constant increase in viral RNA loads until day 7 post infection. In VeroCatSLAM cells, growth was observed from day 1 until day 7, while in LLC-MK2, growth became evident from day 4. Final RNA-load was higher in VeroCatSLAM than in LLC-MK2 (Figure 1A).

To further explore in vitro tropism of FeMV, we repeated the experiment, excluding CRFK cells and measured kinetics of infectious titers of FeMV over 7 days. Vero cells displayed similar kinetics, with slight decrease in infectious titers from 3.17 on day 1 to 2.50 Log_10_ TCID_50_ on day 7 post infection. VeroCatSLAM showed a steep increase in FeMV infectious titer on days 2 and 3 post infection, followed by a slow increase and a peak titer of 5.67 Log_10_ TCID_50_ on day 6 post infection. Interestingly, FeMV replication on LLC-MK2 cells demonstrated constant growth from day 2, reaching the peak titers of 6.59 Log_10_ TCID_50_ on day 7 post infection, providing a 1 log_10_ higher infectious titer than VeroCatSLAM cells (Figure 1B).

### 3.2. SLAMF1 Improved Cellular Susceptibility to FeMV Infection and Replication Kinetics

SLAMF1 is regarded as a primary entry receptor for other morbilliviruses. However, the involvement of SLAMF1 for cell entry of FeMV has not yet been demonstrated. In this study, we compared the ability of FeMV-GT2 to replicate on Vero, VeroDogSLAM, and VeroCatSLAM cells. Infected cells were incubated for 8 days and monitored by microscopical examination daily. At day 8 post infection, supernatant from each well was collected and stored at −80 °C until RNA isolation and qPCR. Remaining cell monolayers infected with FeMV were washed, fixed, and stained using FeMV nucleoprotein rabbit sera.

In contrast to the parental Vero cells, VeroDogSLAM and VeroCatSLAM cells facilitated FeMV replication, as determined by immunofluorescent staining of infected cells and the RNA loads in the supernatant of the infected cells (Figure 2). Furthermore, VeroCatSLAM cells were more permissive than VeroDogSLAM cells, with a mean Ct value of 16.9 compared with 22.4 in the supernatant of VeroDogSLAM cells (Figure 2D).

In contrast to its parental Vero cells (Figure 2A), massive syncytium formation was observed on VeroCatSLAM cells (Figure 2C), whereas small syncytia were observed in VeroDogSLAM cells (Figure 2B). Observed CPEs consisted of multinucleated syncytial giant cells, which is typical for certain paramyxoviruses [35]. Drastic difference in FeMV replication in vitro between parental Vero cells and Vero cells expressing feline SLAMF1 clearly shows the involvement of SLAMF1 in the infection process.

These results support the hypothesis that SLAMF1 (more specifically feline SLAMF1) is involved as a putative receptor for FeMV entry and cell-to-cell fusion in vitro.

### 3.3. In Vivo, the Virus Spreads Systemically Causing Viremia and Transient Fever

Apart from a mild increase in rectal temperature on days 3–5 post infection (Figure 3A), FeMV-GT2 did not induce clinical signs. The cats remained in normal condition and did not lose weight.

A transient and mild increase in WBC counts above normal range (5500–19,500/µL) was observed between days 20 and 49 (Figure 3B). Other blood cell counts (red blood cells and platelets) remain normal along the observation period.

Apart from sporadic increase in aspartate aminotransferase (AST) in 6 out of 15 cats from day 20, all parameters remain normal during the observation period. Urea and creatinine values stayed within the normal range.

Viremia measured by quantitative RT-PCR was usually detectable from day 1 up to day 20 post infection, with all animals being positive at days 3 and 5 post infection. A peak around day 5 (Figure 4A) coincided with the peak of hyperthermia. One cat still had detectable viremia at day 49.

### 3.4. FeMV Shedding in Urine and Swabs from the Upper Respiratory Tract

Viral RNA was detected in the urine from day 7 to 56, with a plateau between days 14 and 49 post infection with a slight peak at day 24 (Figure 4B). The viral RNA load was higher in the urine (above 10^4^ copies/mL) than in the plasma. The plateau of RNA load in the urine was posterior to the peak in the plasma and of higher magnitude. All cats (*n* = 5) had still viral RNA in their urine at day 56 post infection.

Shedding of viral RNA was also observed in swabs from the upper respiratory tract (Figure 4C) at day 5 and 7 post infection. Viral excretion in nasal swabs coincided with the peak of viremia.

### 3.5. Infection Resulted in Seroconversion in All Cats

Apart from two cats with a weakly positive signal, all cats were negative for anti-FeMV-GT2 IFA titers at day (−11). Seroconversion was detected from day 7 and confirmed in subsequent tests with strong positive IFA signals (Figure 5).

When the volume of serum was sufficient, cats were also tested for neutralizing antibodies. Three cats were tested on days 7 and 14 and five cats at day 56. On days 7 and 14, two cats had titers > 1024 and one cat had a titer of 512. On day 56, all cats had titers > 1024.

### 3.6. Histopathological Analysis Revealed Lesions in Kidney and Liver of Infected Cats

Necropsy was performed at day 14, 24, or 56 (5 cats at each time). No gross lesion could be observed.

On day 14, histopathological examination of kidneys revealed multifocal tubular casts in 3 cats out of 5. On days 24 and 56, all cats had multifocal tubular casts. Urinary casts consisted of an accumulation of eosinophilic hyaline or granular material within the tubular lumen, sometimes associated with a degeneration of lining epithelial tubular cells. Lesions of multifocal medullary tubular mineralization were observed in 2 cats on day 24 and 1 cat on day 56. In addition, 1 cat on day 24 and 2 cats on day 56 had lesions of multifocal chronic tubulointerstitial nephritis (Table 2, Figure 6A). Lesions were more pronounced at day 56 than in the previous sampling times.

Immunohistology showed intra-cytoplasmatic localization of FeMV nucleoprotein in tubular epithelial cells. Antigen deposit was mainly at the apical surface of the cells from the cortex (Figure 6B). No staining was observed in the glomeruli or in the transitional epithelium of the renal pelvis.

On day 14, multifocal lymphoplasmacytic portal and interstitial hepatitis as well as hepatocytic hydropic degeneration was observed in all cats. In 4 out of the 5 cats, multifocal portal biliary proliferation and fibrosis was also recorded. In all cats but one, multifocal acute portal hemorrhages were observed. On days 24 and 56, similar lesions were recorded except for portal biliary proliferation and fibrosis lesions which were seen in one cat at day 56 only.

Diffuse activation of lymphoid follicles was noted in the spleen of most animals at all time points (Table 2).

## 4. Discussion

In our in vitro experiments, we evaluated the potential of FeMV-GT2 to grow on different cells lines. Virus isolation and establishment of FeMV assays have been largely hampered by the difficulty to grow FeMV in cell cultures. Using a highly susceptible cell line would facilitate the isolation of new strains, the production of FeMV stocks, and the development of assays such as virus titration and virus neutralization assays.

FeMV-GT2 strain Gordon was previously reported to replicate on kidney cell lines from simian (Vero, LLC-MK2), feline (CRFK), and rodent origin (BHK-21) [31]. LLC-MK2 cells were described as the most susceptible to FeMV-GT2 infection among the tested cell lines, followed by CRFK cells and to a lesser extent Vero, MARC-145, and BHK-21 cells. Our studies confirmed the replication of FeMV-GT2 strain Gordon in LLC-MK2 cells. FeMV-GT2 did not productively infect CRFK cells. Furthermore, in our study, Vero cells did not support an efficient replication of FeMV-GT2. FeMV RNA copy number and virus titer did not increase over a period of seven days (Figure 1A,B). However, Vero cells stably expressing feline SLAMF1 supported FeMV-GT2 growth. From day 1 post infection, both RNA copy numbers and virus titers rapidly increased. FeMV-GT2 replicated, measured by RNA load and virus titer, most productively on VeroCatSLAM cells until the day 5 post infection. In the period between days 5 and 7 post infection, the viral titer on LLC-MK2 cells increased beyond the levels of VeroCatSLAM cells. Marked differences could be observed in the growth of FeMV-GT2 on those two cell lines, primarily because of extensive formation of large multinuclear syncytia during the replication on VeroCatSLAM cells (Figure 2C). This phenotype, readily observed when other morbilliviruses such as Canine distemper virus and Measles virus were grown on Vero cells expressing the respective hosts SLAM [29,35] was, however, not observed on LLC-MK2 cells. It should be mentioned that we initially tested both LLC-MK2 and Vero cells for Nectin-4 and SLAMF1 expression using commercially available antibodies (including feline SLAMF1 and Nectin-4 expression plasmids as controls). None of the two cell lines expressed either of the morbillivirus receptors (data not shown). A possible explanation for the decrease in virus growth on VeroCatSLAM cells after day 4 (Figure 1B) would be a massive destruction of cell monolayer in the initial days post infection, which could not support FeMV replication. Additional optimization steps (MOI, cell density, and harvest timing) on highly susceptible VeroCatSLAM cells are required for reaching higher peak viral titers. Pronounced syncytia formation and marked difference between virus titers between parental Vero cells and VeroCatSLAM cells indicate the important role of SLAM for FeMV-GT2 entry and replication process in vitro.

Furthermore, in our comparative analysis between parental Vero cells, Vero cells stably expressing SLAMF1 from domestic dog (VeroDogSLAM) [29] and VeroCatSLAM cells, we could clearly show the benefit for the virus when cells were stably expressing the domestic cat SLAM receptor, and to a lesser extent the SLAM receptor of domestic dog (Figure 2). This is an indication of evolutionary adaptation of FeMV to use the SLAM receptor from felids and it has been already described in other morbilliviruses [29].

In cats, FeMV is an emerging virus which has been reported worldwide in cats since its first description in 2012 [1]. FeMV was initially isolated from cats with tubulointerstitial nephritis. Conflictual observations have made the association between FeMV infection and renal disease controversial [9,36]. One objective of this study was to evaluate the pathogenesis of FeMV in SPF cats with a specific focus on renal lesions. The FeMV-GT2 Gordon strain was selected because it was isolated from a cat suffering from polyuria-polydipsia syndrome and diagnosed with CKD stage 2 [31,37].

Morbilliviruses are usually responsible for acute infections with severe clinical signs and FeMV is atypical in that respect. Apart from a mild hyperthermia, FeMV-GT2 experimental infection did not induce clinical signs. This is consistent with the fact that FeMV infection in the field has not yet been associated with acute clinical disease. Overall, hematological and biochemical parameters remained in the normal range, consistently with the absence of major clinical signs or extended severe histopathological lesions. We must acknowledge that we did not sample the cats for hematological or biochemical analyses in the first 2 weeks post infection, and so could not detect a possible lymphopenia as usually observed in the acute phase of morbillivirus infection. Nevertheless, our results confirmed some observations in the field. In a small cohort of FeMV-positive field cats, a complete hematological and biochemical analysis did not reveal abnormalities apart from 1 out of 14 FeMV-positive cats with creatinine above the normal range [38]. In contrast, a study involving a larger population of field cats showed a link between seroprevalence and increased creatinine concentrations in the blood [34]. The absence of biochemical abnormalities in our study suggests that markers of renal dysfunction might appear later after infection.

FeMV-GT2 induced a viremia comparable to other morbilliviruses [39]. The decline in viremia after day 7 coincided with the detection of neutralizing antibodies. One cat was still viremic at day 49 suggesting that viremia may persist for a longer period in some infected animals. Viral RNA was also detectable in nasal swabs and coincided with the peak of viremia. This observation may provide a link for the possible transmission of FeMV via an aerosol route as known for all other morbilliviruses [35]. However, due to the low viral loads in the nasal swabs, this route of transmission is probably not a major one. Viral RNA load in the urine was higher than in the blood and was still detectable at a high level in all cats at day 56. Our results of viremia and urinary excretion are consistent with those from cats followed over a long period in the field showing persistent shedding in some animals [6,30]. Our study was stopped at day 56 and did not allow us to assess the duration of excretion of FeMV in the urine. However, urine RNA loads at day 56 suggest that shedding most likely persists beyond 2 months post infection. Interestingly, the RNA loads measured in the urine of experimentally infected cats were in the same magnitude (10^4^–10^5^ RNA copies/mL) as those reported in a field-infected cat [6]. The prevalence of FeMV infection and the sustained urinary excretion may explain the relatively high rate of positive urine highlighted in some surveys [40]. In our in vivo infection experiment, we showed the presence of viral antigen in kidney tubular epithelial cells, demonstrating that the virus can replicate in the feline kidney. The predominant localization at the apical surface of tubular cells is in accordance with virus shedding via urine. In agreement with these results, the presence of FeMV proteins has been demonstrated in renal tubular cells or transitional cells of the renal medulla or pelvis by immunohistochemistry in naturally infected cats [12,13]. In addition, renal epithelial cells and to a much lesser extent urinary bladder cells were shown to be susceptible in vitro to FeMV [31]. In summary, the urinary tract seems to be an important site for viral replication and persistence, although we cannot exclude replication in other organs, as viral RNA was also detected in the spleen and the liver. This assumption is also supported by a recent study showing FeMV antigen localization in the spleen and the urinary bladder of naturally infected cats [19]. In addition, Chaiyasak and colleagues demonstrated deposit of FeMV antigens in the urinary bladder, tracheal, and bronchiolar epithelial cells, lymphocytes and macrophages from spleen and mesenteric lymph node as well as in astro- and oligodendroglia in the brain of two naturally infected cats [19].

All cats seroconverted from day 7 and harbored high antibody titers, as well as high neutralizing antibody titers as shown in a few cats. This rapid induction of high titers of neutralizing antibodies is comparable to other morbilliviruses. Two cats were weakly positive in IFA before experimental infection. This was unexpected for SPF cats and could be explained by false-positive IFA results or a previous exposure to the virus (SPF cats were not tested for FeMV at that time). Importantly, these two cats were negative for viremia at day −11 and responded in the same way to infection as the others, suggesting that they had no solid immunity against FeMV.

Histopathological examination performed at days 14, 24, or 56 showed the presence of a sustained lymphoid activation in the spleen and mild lesions in the liver and the kidneys. These lesions are not expected in SPF cats of that age and may therefore be attributed to FeMV infection. Hepatic lesions were mild but consistently observed in all cats and at all time points. Mononuclear cell infiltration and fibrosis in the portal areas and in the liver parenchyma as well as the presence of FeMV antigen in hepatocytes and monocytes was reported in FeMV-infected cats in the field [8]. FeMV-GT2 infection induced mild kidney lesions which were more marked at day 56 than at days 14 or 24. Multifocal tubular casts were observed in all cats along with multifocal chronic tubulointerstitial nephritis in some cats. These lesions are consistent with those observed in naturally infected cats as well as with the ability of FeMV to infect tubular cells [1,12,13].

Experimental infection with FeMV resulted in viral replication with transient viremia and prolonged urinary excretion. Cats remained healthy and did not show major changes in their hematological and biochemical parameters. However, infection induced mild but consistent lesions in the liver and the kidneys. Obviously, these lesions were too moderate to induce evident organ dysfunction. We did not perform urine analyses. In a retrospective study on field cats, urine analyses in FeMV-infected cats revealed UPC (urine protein:creatinine ratio) higher than in healthy cats [38]. SDS-PAGE of urine proteins showed a reduction in uromodulin and a larger number of low molecular mass protein bands, comparable to cats with chronic kidney disease. Urine analyses results suggested tubular damage and an early renal dysfunction. Questions remain about the evolution of the renal lesions over the longer term in the context of the persistent viral shedding, and after clearance of the virus. Sutummaporn and colleagues have observed a significant difference in the presence of interstitial fibrosis between FeMV-positive and FeMV-negative cats [13]. Interstitial fibrosis is a key component of CKD and the contribution of FeMV to the progression towards CKD needs to be investigated further.

The main limitation of our study is the use of a FeMV strain which was grown in vitro on an LLC-MK2 cell line over 11 passages. Passages were a compulsory step to obtain a stock of virus with a volume and an infectious titer high enough to run an in vivo infection. It cannot be excluded that the adaptation of the virus to the cell line resulted in an attenuation of the virulence of the strain. Sequencing of the inoculated strain should be performed to get a better understanding of the impact of cell passages on the virus. The outcome of experimental infection with the FeMV-GT2 strain grown on LLC-MK2 was consistent with some of the field observations (i.e., absence of reported acute clinical signs, viral loads in urine, persistent excretion in urine, renal lesions, etc.), mainly reported from cats infected with wildtype FeMV genotype 1. This suggests that the passages of the strain did not alter the profile of the pathogenesis of FeMV-GT2 and that both genotypes may share a similar pathogenic profile. Experimental infection with FeMV genotype 1 would be a logical next step. Another limitation is the use of the intravenous route to infect the cats, which is not the natural route. This route was selected for ethical reasons to obtain a consistent infection with a viral inoculum having a moderate infectious titer and reduce the number of animals by limiting the variability of the outcome of the infectious challenge. Additionally, for ethical reasons, no sham-infected cats were included in this study. However, histological examination of liver, spleen, and kidneys from healthy cats of the same colony and age range was generated in another study by the same histopathologist. None of the lesions observed in FeMV-infected cats were present in the tissues of healthy cats.

## 5. Conclusions

This study showed that FeMV shares some features with other morbilliviruses such as the use of the SLAMF1 receptor but displays some differences such as its pathological profile. Further work is needed to investigate the use of other receptors such as nectin-4, which might be of importance in the infection of the urinary tract. The link between FeMV infection and CKD has been controversial. For the first time, the experimental infection of SPF cats showed that the virus can induce lesions in the kidneys, including tubulointerstitial nephritis. The sustained shedding of high viral RNA loads in the urine is compatible with the replication of the virus in the urinary tract. Whether the FeMV-induced lesions of the kidneys can result in CKD will require further investigation, including the follow-up of infected cats over a longer period. A similar study with FeMV genotype 1 is also required, even if field reports suggest a comparable pathogenesis.

## Figures and Tables

**Figure 1 viruses-14-01503-f001:**
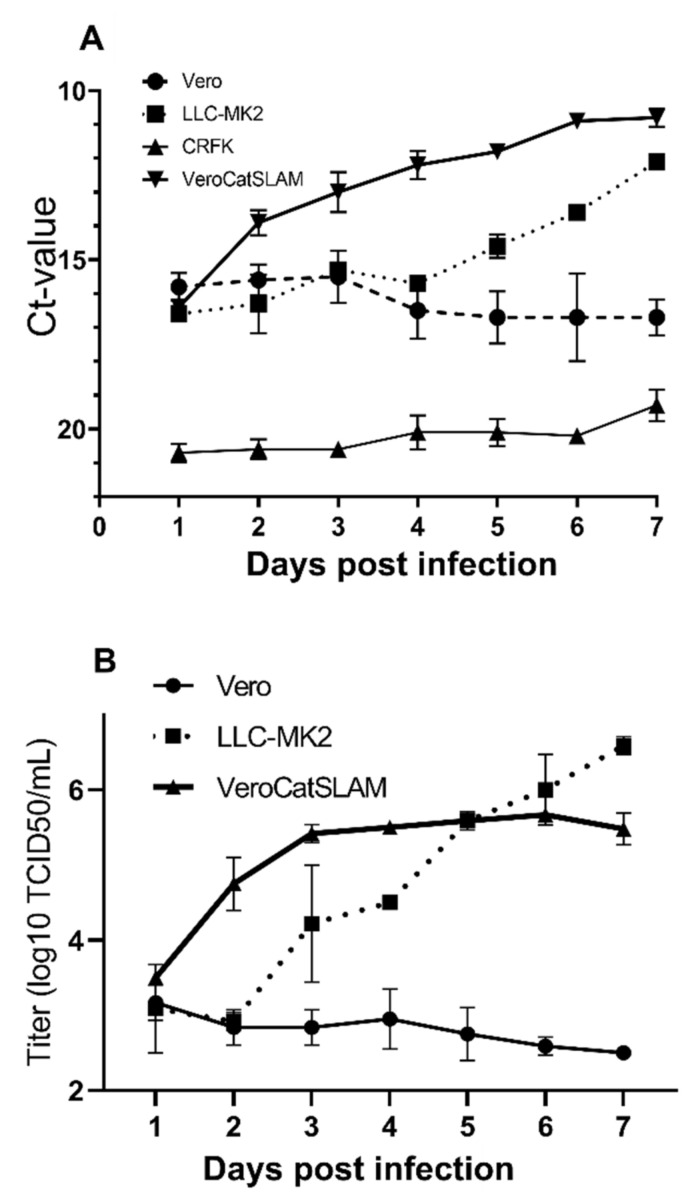
Replication of FeMV-GT2 in vitro. (**A**) Seven-day kinetics of specific FeMV Gordon strain RNA measured by RT-qPCR on LLC-MK2, CRFK, Vero, and VeroCatSLAM cells. Mean values of two distinct experiments and the standard deviation are shown. (**B**) Seven-day kinetics of FeMV replication (TCID_50_) on LLC-MK2, Vero, and VeroCatSLAM cells. Mean values of two distinct experiments and the standard deviation are shown.

**Figure 2 viruses-14-01503-f002:**
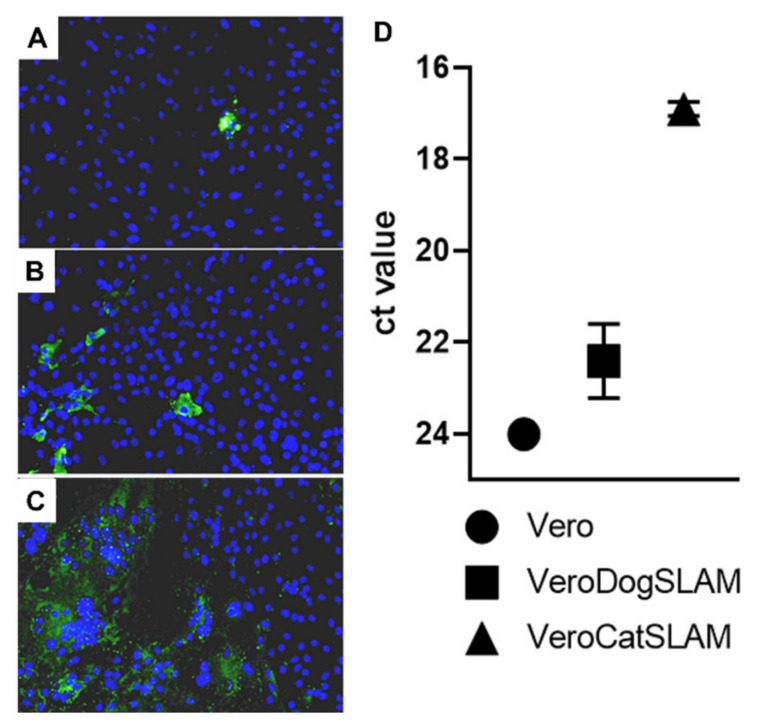
FeMV-GT2 growth 8 days post infection on (**A**) Vero, (**B**) VeroDogSLAM, and (**C**) VeroCatSLAM cells. Green immunofluorescence represents FeMV nucleoprotein signal. Cell nuclei were stained with Hoechst 33342 dye (blue) and FeMV was stained with anti-FeMV-N antibody followed by Alexa 488 conjugated secondary antibody (green). Pictures were taken with a magnification of 100×. (**D**) FeMV RNA loads measured by RT-qPCR. Mean values of two distinct experiments and the standard deviation are shown.

**Figure 3 viruses-14-01503-f003:**
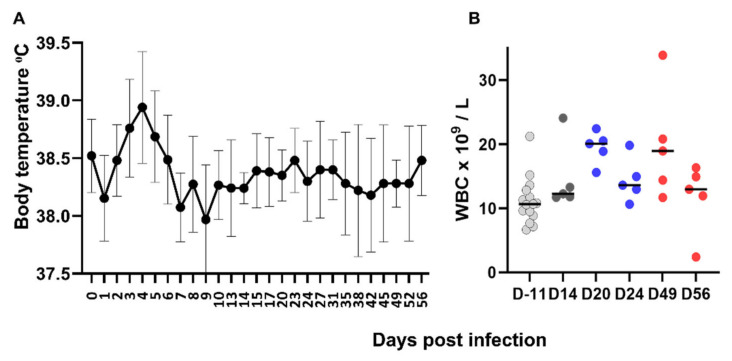
Clinical pathology parameters. (**A**) Rectal temperatures (mean and SD, number of cats: 5, 10, or 15 depending on time point) in cats after experimental infection with FeMV-GT2 on day 0. (**B**) White blood cell counts in cats after experimental infection with FeMV-GT2 on day 0. Individual values and mean values are shown as lines. Point colors refer to the groups and their associated sampling times (Table 1): light grey = all cats, dark grey = group A, blue = group B, red = group C.

**Figure 4 viruses-14-01503-f004:**
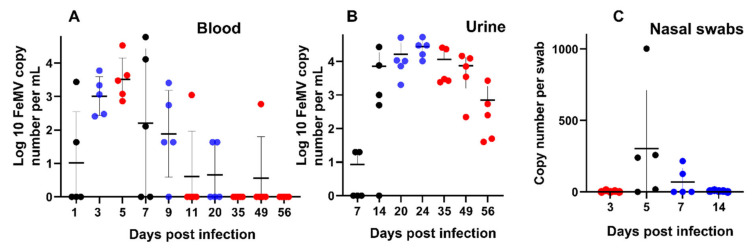
Virus replication and shedding. (**A**) FeMV viremia (FeMV RNA copy numbers/mL of plasma) in cats after experimental infection with FeMV-GT2. (**B**) FeMV excretion in urine (FeMV RNA copy numbers/mL of urine) in cats after experimental infection with FeMV-GT2. (**C**) FeMV excretion in nasal swabs. FeMV-RNA copy number per swab. Cats were infected at day 0. Individual samples are shown as dots. In each column, mean values with standard deviation are shown. Point colors refer to the groups and their associated sampling times (Table 1): black = group A, blue = group B, red = group C.

**Figure 5 viruses-14-01503-f005:**
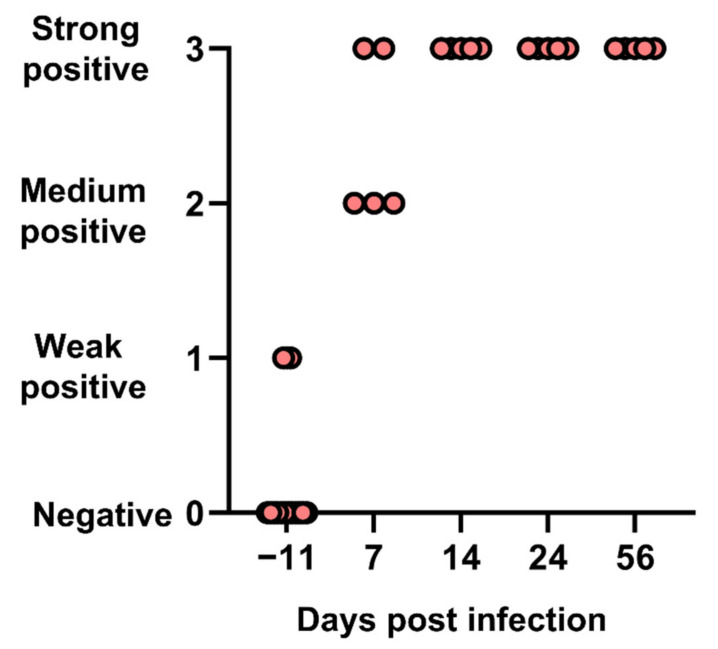
Immunofluorescence antibody (IFA) results in cats after experimental infection with FeMV-GT2 on day 0.

**Figure 6 viruses-14-01503-f006:**
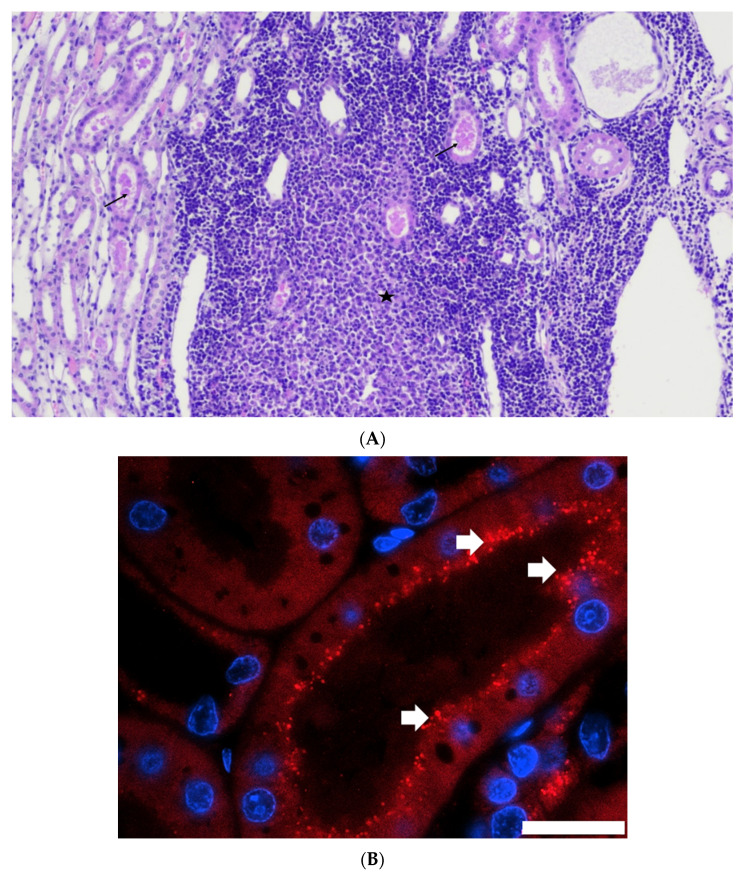
(**A**) Histological section of kidney stained by H&E showing focal tubulointerstitial nephritis (star) and multifocal tubular casts (arrows) in an FeMV-infected cat (day 24). (**B**) Histological section of kidney from an FeMV-infected cat 14 days post infection. Section was stained with a polyclonal rabbit antibody directed against the viral nucleoprotein followed by an AlexaFluor594-conjugated anti-rabbit secondary antibody. Virus-specific signals appear as red dots (see white arrows) and are located at the apical surface of tubular epithelial cells. An FeMV-negative tubule can be seen in the upper left. Cell nuclei were stained with DAPI and are shown in blue. Scale bar represents 20 µm.

**Table 1 viruses-14-01503-t001:** Experimental design of the cat FeMV-GT2 infection study.

Group	Infectious Challenge	IFA	Viremia	Hematology, Biochemistry	Urinary Excretion and Nasal Swabs	Clinical Monitoring	Weighing	Sampling for Histopathological Examination
A (*n* = 5)	d0	d(−11), d7, d14	d(−11), d1, d7	d(−11), d14	d7, d14 d7, d14	d0–d10, d13, d14	d0, d7, d14	d14
B (*n* = 5)		d(−11), d24	d(−11), d3, d9, d20	d(−11), d20, d24	d20, d24 d3	d0–d10, d13, d15, d17, d20, d24	d0, d7, d15, d20, d24	d24
C (*n* = 5)		d(−11), d56	d(−11), d5, d11, d35, d49, d56	d(−11), d49, d56	d35, d49, d56 d5	d0–d10, d13, d15, d17, d20, d23, d27, d31, d35, d38, d42, d45, d49, d52, d56	d0, d7, d15, d20, d27, d35, d42, d49, d56	d56

**Table 2 viruses-14-01503-t002:** Histopathological lesions on days 14, 24, and 56 post infection.

Lesions	Day 14 (Group A)	Day 24 (Group B)	Day 56 (Group C)
Kidney			
Chronic tubulointerstitial nephritis (multifocal)	0/5	1/5	2/5
Tubular casts (multifocal)	3/5	5/5	5/5
Medullary tubular mineralization (multifocal)	0/5	2/5	1/5
Liver			
Lymphoplasmocytic portal and interstitial hepatitis (multifocal)	5/5	3/5	5/5
Portal biliary proliferation and fibrosis (multifocal)	4/5	0/5	1/5
Hepatocytic hydropic degeneration (multifocal)	5/5	5/5	5/5
Acute portal hemorrhages (multifocal)	4/5	2/5	2/5
Spleen			
Lymphoid activation (diffuse)	4/5	5/5	5/5

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
