# Peer review of "In Vitro Growth, Receptor Usage and Pathogenesis of Feline Morbillivirus in the Natural Host"

_viruses, 2022, doi:10.3390/v14071503_

Round 1

Reviewer 1 Report

Review

In vitro growth, receptor usage and pathogenesis of feline morbillivirus in the natural host

By Nikolon et al.,

Submitted to Viruses

The submitted manuscript, a co-operation between scientists from Boehringer-Ingelheim in France and Germany and the Faculty of Veterinary Medicine in Leipzig, analyses the recently detected feline morbillivirus (FeMV). They studied replication in vitro, receptor usage and pathogenesis.

FeMV is a relatively new virus, however it has a worldwide distribution.

FeMV has been associated with tubulointerstitial nephritis (TIN)/chronic kidney disease (CKD), but there were also findings of infected healthy cats without clinical symptoms.

CDK is the most common metabolic disease of domestic cats, affecting 30-40% of cats older than 10 years, a percentage which is higher than the percentage of FeMV-positive cats.

Unclear is also the nature of the virus receptor. Morbilliviruses use signaling lymphocyte activation molecule family member 1 (SLAMF1) or nectin cell adhesion molecule 4 (nectin-4), also known as poliovirus receptor-like 4.

Comment 1

Analyzing the replication of FeMV in vitro, good replication was found on LLC-MK2 cells and VeroCatSLAM cells, i.e., monkey Vero cells expressing the cat SLAM receptor, but not on Vero or CRFK cells. In a previous publication the authors however described good replication on Vero and CRFK cells. Did the virus or the receptor usage change over time?

Comment 2

In the case of the measles virus in addition to SLAM1 and nectin-4 CD46 is shown to act as receptor. Is there evidence in the case of FeMV?

Comment 3

The viremia declined after day 7, this coincided with the detection of neutralizing antibodies.

Is the early induction of neutralizing antibodies exceptionally early in comparison with other viruses? For comparison, in the case of HIV-1 antibodies are detected after 3 to 12 weeks. The authors write that the rapid induction is typical for morbilliviruses. Please discuss.

Comment 4

Despite the decline of the viremia, the virus shedding in the urine continues. Does this mean that the virus-producing cells are not accessible by the immune system? Please discuss.

Comment 5

The main finding of this work is the detection of mild lesions in the liver and the kidney. Since the authors used SPF animals, the contribution of a second agent can be excluded. It is a pity that the experiment was stopped after 56 days. The question is, whether the kidney lesions observed after 56 days could develop into the usually described tubulointerstitial nephritis (TIN)/chronic kidney disease (CKD) in diseased animals?

Comment 6

Most of the Figures are too large and should be reduced.

Comment 7:

Figure 9. Please show in parallel a normal kidney section and indicate the focal tubulointerstitial nephritis and multifocal tubular casts in the affected kidney.

Comment 8

How do the authors explain the fact that all other morbilliviruses including measles virus, canine distemper virus (CDV), phocine distemper virus (PDV), peste des petits ruminants virus, rinderpest virus, and the porcine morbillivirus induce severe and fatal diseases, but FeMV not?

Comment 9

Morbilliviruses cause profound immune suppression. Was there evidence of immunosuppression in this experiment, in other experiments? Obviously FeMV can infect lymphocytes and lymphocytes carry the SLAM1 receptor. However, most morbilliviruses have been shown to act indirectly, inhibiting lymphocyte proliferation by their fusion and the hemagglutinin proteins.

Minor

Line 92: please explain MeV

Line 108: please add for a better understanding, that the FeMV-GT2-Gordon strain was isolated from a cat called Gordon.

In summary, the manuscript is original and the results are interpreted appropriately.

The article written in an appropriate way, the methods are described with sufficient details. Some passages are too lengthy.

The work provides an advance towards the current knowledge and the English language is appropriate and understandable.

Author Response

Analyzing the replication of FeMV in vitro, good replication was found on LLC-MK2 cells and VeroCatSLAM cells, i.e., monkey Vero cells expressing the cat SLAM receptor, but not on Vero or CRFK cells. In a previous publication the authors however described good replication on Vero and CRFK cells. Did the virus or the receptor usage change over time?

Answer: This is very valid question, thank you. We will try to give more details which would hopefully clarify the confusion present in the literature. In our study, we analyzed in vitro growth of FeMV GT2, strain Gordon. It could be that different strain have somewhat different in vitro characteristics.

The isolation of FeMV is exceptionally challenging. To our knowledge, only two FeMV virus isolations where virus stocks were produced and successfully titrated.

In Woo et al. 2012, Vero E6 and CRFK cells were used to isolate the virus. Incubation for each passage was 2-3 weeks. For the virus isolation 16 continuous passages were needed, to obtain virus stock. The second successful cultivation was done at Veterinary faculty at University of Leipzig, where our material is produced. We were for the first time able to successfully grow the virus and to establish titration assays for quantification of infectious titers.

In general virus cultivation and virus stock preparation was challenging in the past. Some other groups (I.e. Sakaguchi et al 2015, Donato et al. 2019) used blindly sample material incubated on various cells and tested virus replication by RT-qPCR measuring RNA loads. This gave impression that e.g., VERO cells show good performance in supporting the virus growth. Our identification of LLC-MK2 cells as well as VERO cells expressing cat SLAM (VCS) receptor significantly improved virus growth and allowed us to establish virus titration based on visible syncytia formation (VCS).

Comment 2

In the case of the measles virus in addition to SLAM1 and nectin-4 CD46 is shown to act as receptor. Is there evidence in the case of FeMV?

Answer: In case of measles virus and canine distemper virus, CD46 receptor could be used only by vaccine strains e.g., Edmondston strain (MeV) or Onderstepoort and Lederle strain of CDV. These viruses, through the extensive passaging on SLAM negative cells gained the ability to more efficiently enter via CD46. Wild type strains of morbilliviruses do not efficiently use CD46 for their entry, nor can use CD-46 expressing cells for replication in vitro.

Comment 3

The viremia declined after day 7, this coincided with the detection of neutralizing antibodies.

Is the early induction of neutralizing antibodies exceptionally early in comparison with other viruses? For comparison, in the case of HIV-1 antibodies are detected after 3 to 12 weeks. The authors write that the rapid induction is typical for morbilliviruses. Please discuss.

Answer: This statement is based on our experience with canine distemper virus (CDV), another morbillivirus. We agree with the reviewer’s comment on neutralizing antibodies against lentivirus, and more generally retroviruses. Detection of neutralizing antibodies can be more rapid with some viruses than others. The objective was not to say that seroconversion is more rapid with morbilliviruses than other viruses but rather, that this kinetics looks similar to what is observed with other morbilliviruses like CDV.

Comment 4

Despite the decline of the viremia, the virus shedding in the urine continues. Does this mean that the virus-producing cells are not accessible by the immune system? Please discuss.

Answer: Thank you for this question, which leads to the core of understanding of FeMV pathogenesis. The hallmark of morbillivirus infection is initial virus spread systemically in lymphatic tissues, peaking 5-9 days post infection. During that time, apart from fever, there are no specific signs of disease (except in some highly susceptible host like CDV in mustelids)

In the second phase of infection, the viruses are usually cleared from systemic circulation, but replicate in various tissues (Lungs, skin, LN, spleen, CNS) and most of the pathologies observed are associated with morbillivirus replication in periphery. In those tissues the virus replicates mostly without extracellular phase and is most likely transmitted by cell-to-cell fusion. In these conditions, the virus is less exposed to the immune system.

This fits together with the FeMV pathogenesis which we could observe in our study, with the difference that FeMV in the second phase accumulates in the urinary tract and is able to persist very long (according to our study, more than 56 days). Although the presence of CDV has been demonstrated in urine of infected canids and mustelids, the long persistence of FeMV in urinary tract is very unique for a morbillivirus.

Comment 5

The main finding of this work is the detection of mild lesions in the liver and the kidney. Since the authors used SPF animals, the contribution of a second agent can be excluded. It is a pity that the experiment was stopped after 56 days. The question is, whether the kidney lesions observed after 56 days could develop into the usually described tubulointerstitial nephritis (TIN)/chronic kidney disease (CKD) in diseased animals?

Answer: It is indeed a limitation of our study. Our primary objective was to study the impact of infection on kidneys. We confirmed the presence of moderate lesions consistently with field reports. Whether these lesions can lead to CKD will require additional studies. The pathogenesis of feline CKD is not fully understood. It is multifactorial. Whether mild injuries associated with infections like FeMV can contribute to the evolution towards CKD remains to be investigated.

Comment 6

Most of the Figures are too large and should be reduced.

Answer: we will address this comment with the editor in the context of the finalization of the manuscript. The size of the figures can be modified very easily

Comment 7:

Figure 9. Please show in parallel a normal kidney section and indicate the focal tubulointerstitial nephritis and multifocal tubular casts in the affected kidney.

Answer: focal TIN and casts are shown (arrows and a star were added to indicate the lesions). We don’t think that adding a normal picture would be of great added value. Normal renal tissue can be seen on the left of the photo. As indicated to another reviewer, we did not include sham infected cats for ethical reasons. However, we have histological data from healthy cats from the same colony and age range. None of the lesions observed in FeMV infected cats were present in those healthy cats.

Comment 8

How do the authors explain the fact that all other morbilliviruses including measles virus, canine distemper virus (CDV), phocine distemper virus (PDV), peste des petits ruminants virus, rinderpest virus, and the porcine morbillivirus induce severe and fatal diseases, but FeMV not?

Answer: indeed, morbilliviruses are usually responsible for acute severe diseases. To our knowledge, there has been no report of acute disease and mortality associated to FeMV infection. Furthermore, in our study, we didn’t observe serious acute clinical signs upon experimental infection, confirming field observations. It is most likely a specificity of this virus which makes it a rather atypical morbillivirus.

Comment 9

Morbilliviruses cause profound immune suppression. Was there evidence of immunosuppression in this experiment, in other experiments? Obviously FeMV can infect lymphocytes and lymphocytes carry the SLAM1 receptor. However, most morbilliviruses have been shown to act indirectly, inhibiting lymphocyte proliferation by their fusion and the hemagglutinin proteins.

Answer: we didn’t perform immunological tests to detect an hypothetical FeMV-induced immunosuppression. It is indeed an interesting aspect of the infection to be explored.

Minor

Line 92: please explain MeV

Answer: done

Line 108: please add for a better understanding, that the FeMV-GT2-Gordon strain was isolated from a cat called Gordon.

Answer: done

In summary, the manuscript is original and the results are interpreted appropriately.

The article written in an appropriate way, the methods are described with sufficient details. Some passages are too lengthy.

The work provides an advance towards the current knowledge and the English language is appropriate and understandable.

Answer: we thank the reviewer for these positive comments. We are aware of the limitations of our study but hope it contributes to the knowledge of FeMV pathogenesis and will trigger additional investigations to address the remaining questions.

Reviewer 2 Report

In this study by Nikolin et al, the authors assess the replication of feline morbillivirus (FeMV) in different cell lines, investigate receptor usage and perform an in vivo infection experiment in cats to determine various parameters of the clinical infection under controlled conditions. Since the first report of feline morbillivirus in Hong Kong in 2012, many clinical studies incorporating some degree of limited phylogenetic studies have demonstrated the widespread prevalence of FeMV in cats, characterized a range of strains clustering with two genotypes and have reported an association with chronic kidney disease (CKD). It is the latter point which remains a key research topic and one which is important to clarify.

The manuscript is well written, and the introduction in particular provides a clear overview of the current status of the FeMV research field. The data presented on the use of feline CD150 is convincing but clearly it would be useful in future research studies to determine binding affinities for the FeMV H – feline/canine CD150 interaction. The in vivo infection experiment is important for the FeMV research field as it clearly demonstrates the long term persistence of virus (Figure 6) in the absence of overt clinical symptoms, confirming observations which have been previously suggested by clinical case reports. This also provides additional avenues of research connected to the molecular biology and virus-host interactions to determine why the clinical course of infection is so different from that of the classical morbilliviruses.

The clear outline of the limitations of this study in lines 502-508 is welcome as this provides clear avenues for future studies with respect to (i) more comprehensive sequence analysis of FeMV in an infected animal in comparison to the early and high passage virus isolates derived from that same animal (ii) longer term studies in the in vivo cat model to better determine the postulated link to CKD.

I only have minor comments:

  • Is there a control avaliable (in vitro transcribed RNA from a cloned amplicon covering the target region) for the RT-qPCR assay to determine copy number? i.e., how was the copy number calculated for the data presented in Figure 5?
  • The data regarding the high virus titers obtained from FeMV infection of LLC-MK2 are intriguing. Is it known if these cells express Nectin 4?
  • Line 284, Figure 2. I would describe the foci of infected VeroDogSLAM cells as ‘small syncytia’ or ‘small foci of 2-3 fused cells’ rather than ‘moderate syncytia’ unless a more convincing image is avaliable to show larger syncytia i.e., something in between the images in (a) and (c).
  • I would combine Figures 3, 4 and 5 into one figure (Figure 3 a, b, c) since they are showing different aspects of the clinical infection.
  • Similarly, I would combine Figures 6 and 7 since they both show virus load data during the clinical infection.
  • I would also combine Figure 9 and 10 since they show the histology/virus distribution and are thus complementary.
  • It is important to state in the conclusions that analogous studies need to be performed with genotype 1 FeMV isolates to assess if there are genotype/strain differences with respect to disease severity.
  • Additional minor proof reading is required throughout the manuscript to correct minor mistakes/missing words.

Author Response

I only have minor comments:

  • Is there a control avaliable (in vitro transcribed RNA from a cloned amplicon covering the target region) for the RT-qPCR assay to determine copy number? i.e., how was the copy number calculated for the data presented in Figure 5?

Answer: the copy number was calculated by using a reference plasmid with calibrated copy number.

  • The data regarding the high virus titers obtained from FeMV infection of LLC-MK2 are intriguing. Is it known if these cells express Nectin 4?

Answer: Indeed, we tested the expression of SLAM and Nectin-4 in LLC-MK2 cells and VERO. None of the cells showed expression of the potential FeMV receptors. We added this information to the discussion section.

  • Line 284, Figure 2. I would describe the foci of infected VeroDogSLAM cells as ‘small syncytia’ or ‘small foci of 2-3 fused cells’ rather than ‘moderate syncytia’ unless a more convincing image is avaliable to show larger syncytia i.e., something in between the images in (a) and (c).

Answer: The authors agree with the proposal. Done.

  • I would combine Figures 3, 4 and 5 into one figure (Figure 3 a, b, c) since they are showing different aspects of the clinical infection.

Answer: we agree with the proposal. Done

  • Similarly, I would combine Figures 6 and 7 since they both show virus load data during the clinical infection.

Answer: done

  • I would also combine Figure 9 and 10 since they show the histology/virus distribution and are thus complementary.

Answer: done

  • It is important to state in the conclusions that analogous studies need to be performed with genotype 1 FeMV isolates to assess if there are genotype/strain differences with respect to disease severity.

Answer: we agree with the comment. Done.

  • Additional minor proof reading is required throughout the manuscript to correct minor mistakes/missing words.

Answer: done

We thank the reviewer for the positive comments. We hope that this study contributed to the knowledge on FeMV and will trigger additional investigations to address the remaining questions on its role and impact on cat health.

Reviewer 3 Report

The authors have attempted to answer numerous fundamental questions regarding the receptor usage and pathogenesis of FeMV; knowledge gaps that have existed since the discovery of the virus in 2012.

Unfortunately, the manuscript as written is potentially misleading and cannot be published without additional data.

Fortunately the final publication of this special issues of Viruses, on feline viruses, has been delayed until October 2022, which gives the authors time to obtain this data, and provide clarity in the manuscript.

The following additional data must be obtained and reported prior to resubmission:

  1. The whole genome sequence of the virus (obtained from passage 11 in monkey cells) used to infect the cats must be obtained, compared to the original FeMV-GT2 clinical isolate and reported – it is clear that whatever is growing in the monkey cells must be an adapted virus (not wildtype FeMV-GT2), and so reporting how different it is from the clinical isolate is vital
  2. Incongruent findings in the three key methods (ie. replication of FeMV in various cell lines, catSLAM improving replication, and in vivo biology) require clarification
    1. If catSLAM is required for viral entry, why does the virus grow so well in LLC-MK2 cells (Results section 3.1 and Figure 1A, B) such that it was the only cell line you could grow up titers need for infections
    2. Figures 1A and 2D are conflicting – in 1A the mean Ct value (on d7) in VeroCatSLAM is 11, whereas the mean Ct in Figure 2D (day 8) is 17. In
    3. If catSLAM is required for viral entry, and the virus is identified in cat kidneys after experimental infection, why doesn’t the virus replicate in CRFK cells. Virus isolations from the urine would aid here
    4. Figure 2B also appears to show that this virus also uses DogSLAM
  3. The serology requires improvement. It is unclear why virus neutralisation assays were not performed to confirm and accurately quantify seroconversion in all of the animals. it is unclear how the semiquantitative data reported in Figure 8 was obtained – numerical data around what constituted weak, medium and strong positive is required. Additionally, the authors must justify why two seropositive cats were included in the experimental infections.
  4. Additionally the lack of contemporaneously sham infected cats means that the post-mortem histopathological changes (where IHC is lacking) cannot be definitively attributed to FeMV.
  5. Significantly more information about the animal experiments (outlined below).

The aforementioned key concerns require addressing. More minor comments are included below.

Introduction:

  • Page 1, line 39 – samples of what?
  • Line 57 – it is inaccurate to say that the cats in the study reference [4], had signs of CKD – some only had lower urinary tract signs – please reword
  • Line 60 – for reference [13] can you please provide slightly more information here
  • Line 71 – change to “cats”
  • Line 78 – please add the word “and” before “ii)”

Materials and Methods:

  • Can you please clarify how you confirmed that these cells were stably expressing DogSLAM and catSLAM
  • Line 129 - Can you clarify how the anti-FeMV-N rabbit serum was derived, and data attesting to its specificity
  • Regarding the RT-PCR – please clarify the quantities primers and probe used in your reactions, the cycling conditions, and size of the target amplicon. Also, what positive control material was used?
  • Please use an uppercase L when writing uL (eg. line 151, 152, 154, 216)
  • Line 157 – can you clarify what specific CPE- particularly given that FeMV infection has not reliably created CPE
  • Line 171 – please add the word “to” before “supernatant” here
  • Please add significantly more detail about the animal experiments – how was the IV injection (challenge) given – via an IV catheter to ensure it all went IV? While sedated? how was blood sampled, how was urine sampled, how were nasal swabs collected? how was body temperature measured (and when relative to sampling or other procedures), were cats sedated for any of this sampling – if so, with what, how were the cats ultimately euthanised, and how long was the delay between euthanasia and post-mortem examination / collection of tissues for histopathology
  • Additionally, how was nucleic acid extracted from blood? (whole blood vs PMBCs) urine? and nasal swabs?
  • Additionally – how were cats housed? Cohoused in groups or individually? How frequently were cats observed for clinical signs? What clinical signs were staff looking for?
  • What was the delay to performing haematology and biochemical analyses? since it appears that samples were sent to an external laboratory?
  • Abbreviations should be introduced the first time they are used, and then used throughout thereafter eg. IFA, DAPI
  • Line 190 - Was histopathology performed by a board certified veterinary pathologist?

Results:

  • Weighing is the cats is mentioned in the methods section but body weight data is not included in the results – did they lose weight? This might be an indication of illness although no outward signs were observed
  • Table 2 and line 355 – can you please clarify what type of tubular casts were present? Since some types of casts (eg. hyaline) are not pathologic
  • What is meant by lymphoid activation in the spleen? Table 2 and line 370
  • Line 220 – presumably you are referring to cats persistently infected with FeMV-GT2 – please clarify
  • Line 222 – please provide more information regarding how you performed your semi-quantitation, and why this seemed to be prioritised over the more appropriate neutralisation assays
  • Line 301 – can the reference interval be included in the text or on Figure 4
  • Bar charts are not an appropriate way to display the data in 5, 6 and 7 – box and whisker plots (or mean +/- SD/SEM) could be used with your overlay of dot plots, or just dot-plots given the low number of animals at each time point
  • Additionally, the numbers / proportions of cats that were negative for viral RNA in their blood, urine, and nasal swabs should be made clear
  • Line 327 – please clarify that this is only 5 cats
  • Line 346 – given the superiority of virus neutralisation assays for serology please include the data here (in addition to reporting why neutralisations were not done for all serologic samples)
  • Figure 9 – at what day was this cat sacrificed/ HP performed?

Discussion:

  • How do you reconcile the growth of FeMV in LLC-MK2 cells when presumably they don’t express catSLAM – this should be clarified?
  • How do you reconcile the “weakly positive signal” in IFA in two cats at baseline. Although they are SPF cats, it is possible that they could have had prior exposure to FeMV, since it has not been an agent SPF cats are screened for historically. Why were they then not excluded from the infections
  • Line 407 – remove the period after “[29,35]”
  • Line 413-418 – can you expand upon this – do other morbilliviruses use SLAM from multiple species, could this just be adaptation to cell culture?
  • Line 446 – the viral loads in nasal swabs were very low – can you comment on this in the context of possible aerosol transmission
  • A significant limitation is that the pathogenesis of the virus used to infect the cats (adapted to monkey cells) cannot be assumed to be the same as FeMV-GT 2, which in turn cannot be assumed to be the same as FeMV-GT1, which appears to be the more prevalence circulating strain based on existing data. WGS data must be provided for the virus used to perform the infections, and then the difference between this and wildtype strains must be clarified this as a limitation.
  • You mention the low level of SLAM sequence conservation between species (line 100), but then don’t discuss (just mention) the apparent use of dogSLAM by the virus. This requires clarification (eg. in the discussion lines 413-416).

Author Response

The following additional data must be obtained and reported prior to resubmission:

  1. The whole genome sequence of the virus (obtained from passage 11 in monkey cells) used to infect the cats must be obtained, compared to the original FeMV-GT2 clinical isolate and reported – it is clear that whatever is growing in the monkey cells must be an adapted virus (not wildtype FeMV-GT2), and so reporting how different it is from the clinical isolate is vital

Answer: We agree with the reviewer’s comment. Sequencing the viral genome is likely to show some mutations in various genes, as shown when passaging other morbilliviruses like canine distemper virus, rinderpest virus, peste-des-petits-ruminants virus or measles virus (Eloiflin 2019 Viruses; Wu et al. 2016 Viruses Genes; Baron et al. 1996 J Gen Virol). However, the interpretation of these differences is difficult from a functional standpoint and does usually not bring much added value. The mechanism of attenuation by cell passages is not fully understood (Liu et al. 2016 Comp Immunol Microbiol).

The passages on monkey cells were motivated by the need to obtain a stock of virus with an infectious titer high enough to be able to infect cats. They were kept to a minimum to try and limit the attenuation of the strain. It was a compulsory step as it was the only mean to generate virus stocks for clinical use at the time when VeroSLAM cells were still not tested.

The experimental infection of cats with this strain resulted in changes that are consistent with the various reports from field infection, suggesting that this adaptation may have reduced the virulence of the Gordon strain without changing its overall pathogenesis.

Comments were added in the limitations of the study.

  1. Incongruent findings in the three key methods (ie. replication of FeMV in various cell lines, catSLAM improving replication, and in vivo biology) require clarification
    1. If catSLAM is required for viral entry, why does the virus grow so well in LLC-MK2 cells (Results section 3.1 and Figure 1A, B) such that it was the only cell line you could grow up titers need for infections

Answer: Based on our results, the presence of CatSLAM significantly improves the replication of the virus used in our study.

However, we do not attempt to claim that cat SLAM is absolutely required for infection, nor that FeMV exclusively use SLAM for cell entry. It has been shown with other morbilliviruses that they can infect and grow in cells that do not express SLAM or even Nectin-4. As an example, some morbillivirus strains can use CD46 for entry (Nielsen et al. 2001 Arch Virol).

    1. Figures 1A and 2D are conflicting – in 1A the mean Ct value (on d7) in VeroCatSLAM is 11, whereas the mean Ct in Figure 2D (day 8) is 17. In

Answer: The authors thank the reviewer for this question. We will try to give further explanation.

Figure 1 demonstrate the growth curve of FeMV strain Gordon measured by sampling the total content of the well for the analysis (cells were scraped and resuspended in existing media in the well). The methodology for the next experiment (Figure 2) was different. In this case, we cultivated the virus for 8 days, followed by sampling supernatant in the well, while the monolayers were acetone fixed and FeMV stained (this was the source of IFA photos –Fig 2.a-c. Virus loads in the supernatants indicate the amount of cell free virus fraction and are regularly lower than in a cell fraction, which explains the difference in RNA copies of the virus. This is already described in the methods section.

    1. If catSLAM is required for viral entry, and the virus is identified in cat kidneys after experimental infection, why doesn’t the virus replicate in CRFK cells. Virus isolations from the urine would aid here

Answer: Morbilliviruses behave and replicate very differently in continuous cells lines and in vivo in their host. Furthermore, CRFK is a clonal cell line and its phenotype (mesenchymal or epithelial) is not fully understood. Determination of target cells in the kidneys for FeMV will require additional studies.

    1. Figure 2B also appears to show that this virus also uses DogSLAM

Answer: Thank you for this observation. The ability of many different morbilliviruses (e.g. CDV and measles virus) to use SLAM receptors from other species is well documented (e.g. Fukuhara et al 2019 and Nikolin et al. 2012). In line with those reports, we detected higher affinity of FeMV for cat SLAM (real host) than to dog SLAM.

  1. The serology requires improvement. It is unclear why virus neutralisation assays were not performed to confirm and accurately quantify seroconversion in all of the animals. it is unclear how the semiquantitative data reported in Figure 8 was obtained – numerical data around what constituted weak, medium and strong positive is required. Additionally, the authors must justify why two seropositive cats were included in the experimental infections.

Answer: As explained in the manuscript, we had not enough serum to run virus neutralization test (VNT) for all cats. VNT was not yet available when we initiated the study. It was added afterwards for those cats for which we had enough serum available.

Our results are consistent and homogeneous enough however to give a good overview of the seroconversion and appearance of neutralizing antibodies.

We had indeed two weakly positive cats before experimental infection. This came as a surprise to us because these are SPF cats. We did not eliminate those cats because cats are precious and difficult to obtain nowadays, and we did not want to reduce the power of the study.

Importantly, these two cats were negative for viremia at day -11 and responded in the same way to infection as the others, suggesting that they had no solid immunity against FeMV.

Comment was added in the discussion.

  1. Additionally the lack of contemporaneously sham infected cats means that the post-mortem histopathological changes (where IHC is lacking) cannot be definitively attributed to FeMV.

Answer: for ethical reasons, we did not include sham infected cats. In another study, we have histopathological data from two non-infected cats of the same age from the same colony and examined by the same histopathologist. The histological examination of those two cats showed normal kidney, liver and spleen tissues. None of the observations reported in our study were observed in those cats. This comment will be added to the manuscript. Regarding the origin of the lesions reported in our study, they are most likely cause by FeMV for the following reasons:

  • The cats are SPF cats that were kept in BSL-2 environment, preventing infection by other agents
  • Another FeMV challenge study was performed for the evaluation of a therapy and yielded the same observations.

A comment will be added in the limitations of the study at the end of the discussion.

  1. Significantly more information about the animal experiments (outlined below).

The aforementioned key concerns require addressing. More minor comments are included below.

Introduction:

  • Page 1, line 39 – samples of what?

Answer: Thanks for spotting this, we added the change to the manuscript.

  • Line 57 – it is inaccurate to say that the cats in the study reference [4], had signs of CKD – some only had lower urinary tract signs – please reword

Answer: was changed to “urinary tract disease”

  • Line 60 – for reference [13] can you please provide slightly more information here

Answer: Done. “… a significant association with between the presence of FeMV antigen and morphologic tubular and interstitial alterations in cat kidneys”

  • Line 71 – change to “cats”

Answer: Thanks for spotting this typo, we added the change to the manuscript.

  • Line 78 – please add the word “and” before “ii)”

  Answer: Thanks for spotting this typo, we added the change to the manuscript.

Materials and Methods:

  • Can you please clarify how you confirmed that these cells were stably expressing DogSLAM and catSLAM

Answer: The cells used in this study we acquired from FU Berlin and their development has been already described (citation 29. in the manuscript). Furthermore, cells were constantly under selective pressure with Zeocin 0.5 mg/ml to ensure strict selection of SLAMF1 positive cells. We provided more details in the manuscript (lines 120-125).

  • Line 129 - Can you clarify how the anti-FeMV-N rabbit serum was derived, and data attesting to its specificity

Answer: Recombinant FeMV N protein was produced in E. coli and used to immunize rabbits. Antiserum was shown to be specific of FeMV N protein. In particular, the antiserum does not react with other common feline viruses or Canine Distemper virus in IF assay.

  • Regarding the RT-PCR – please clarify the quantities primers and probe used in your reactions, the cycling conditions, and size of the target amplicon. Also, what positive control material was used?

Answer: the quantities of primers and probe are presented in the table below:

reagent

vol for 1 reaction (µl)

C finale

eau Nuclease free

4

2X RT PCR buffer

12,5

1X

primer forward 10µM

1

400nM

primer reverse 10µM

1

400nM

probeFeMV_N  10µM

0,5

200nM

25X RT PCR enzyme

1

vol total :

20

+ RNA (matrix)

5µl

The cycling conditions are described in the table below:

cycle

repeat

time

set point

PCR/melt

1

1

10 min

45°C

2

1

10 min

95°C

3

40

15 sec

95°C

45 sec

56°C

45 sec

60°C

Real Time

The target amplicon is 200 nucleotides long.

As positive control, we used a reference plasmid.

  • Please use an uppercase L when writing uL (eg. line 151, 152, 154, 216)

Answer: Thanks for spotting this typo, we added the change to the manuscript.

  • Line 157 – can you clarify what specific CPE- particularly given that FeMV infection has not reliably created CPE

Answer: We added necessary clarification to the manuscript.

  • Line 171 – please add the word “to” before “supernatant” here

Answer: Thanks for spotting this typo, we added the change to the manuscript.

  • Please add significantly more detail about the animal experiments – how was the IV injection (challenge) given – via an IV catheter to ensure it all went IV? While sedated? how was blood sampled, how was urine sampled, how were nasal swabs collected? how was body temperature measured (and when relative to sampling or other procedures), were cats sedated for any of this sampling – if so, with what, how were the cats ultimately euthanised, and how long was the delay between euthanasia and post-mortem examination / collection of tissues for histopathology

Answer: For the FeMV challenge, cats were sedated and a catheter was used to ensure that the inoculum was delivered intra-venously. Urine was collected by cystocentesis. Blood was collected from the jugular vein. Animals were sedated only if necessary, by using Domitor for sedation and Antisedan for awakening. Tissues were collected immediately after euthanasia.

  • Additionally, how was nucleic acid extracted from blood? (whole blood vs PMBCs) urine? and nasal swabs?

Answer: Nucleic acid extraction is described in chapter 2.3 of the manuscript. For blood, RT-PCR was run using plasma. This will be added in the manuscript (section 2.3)

  • Additionally – how were cats housed? Cohoused in groups or individually? How frequently were cats observed for clinical signs? What clinical signs were staff looking for?

Answer: cats were housed together in groups. Clinical examination was done once a day. The clinical signs the staff was looking for (temperature, weight loss, general condition, dehydration, vomiting, diarrhea, abdominal pain, nasal or ocular discharge, buccal ulcers and any other clinical abnormality) are added in the manuscript (material and methods).

  • What was the delay to performing haematology and biochemical analyses? since it appears that samples were sent to an external laboratory?

Answer: samples were sent to the external lab in the morning and were analyzed in the afternoon.

  • Abbreviations should be introduced the first time they are used, and then used throughout thereafter eg. IFA, DAPI

Answer: done

  • Line 190 - Was histopathology performed by a board certified veterinary pathologist?

Answer: The histopathology was performed by a diplomate of the European College of Veterinary Pathologists at Vet Diagnostics lab. Interestingly, and not written in the manuscript, some histopathological examinations were repeated at the University of Leipzig and led to the same conclusions.

Results:

  • Weighing is the cats is mentioned in the methods section but body weight data is not included in the results – did they lose weight? This might be an indication of illness although no outward signs were observed

Answer: the cats did not lose weight, consistently with their general condition which remained good. A sentence was added in the manuscript.

  • Table 2 and line 355 – can you please clarify what type of tubular casts were present? Since some types of casts (eg. hyaline) are not pathologic

Answer: The casts consisted of an accumulation of eosinophilic hyaline or granular material within the tubular lumen, sometimes associated with a degeneration of lining epithelial tubular cells.   

A sentence was added in the results section.

  • What is meant by lymphoid activation in the spleen? Table 2 and line 370

Answer: diffuse activation of lymphoid follicles was observed. Precision added to the manuscript.

  • Line 220 – presumably you are referring to cats persistently infected with FeMV-GT2 – please clarify

Answer: yes. Done

  • Line 222 – please provide more information regarding how you performed your semi-quantitation, and why this seemed to be prioritised over the more appropriate neutralisation assays

Answer: Thanks for the question. At the time when the study was initiated, the virus neutralization assay was not yet available. It was established in our lab only later, during the course of the study.

Semi-quantitative IFA response was elevated using different dilutions from our positive control sample (serum from FeMV-2 persistently infected cat ‘Gordon’). Low, moderate and high was defined by the intensity of the FeMV-specific fluorescence signal in comparison to the aforementioned positive control serum (and to the signal intensity when using the FeMV-N antibody). Reactivity of the positive control was defined as ‘high’.

  • Line 301 – can the reference interval be included in the text or on Figure 4

Answer: done. (5,500 to 19,500/µl) was added in the text

  • Bar charts are not an appropriate way to display the data in 5, 6 and 7 – box and whisker plots (or mean +/- SD/SEM) could be used with your overlay of dot plots, or just dot-plots given the low number of animals at each time point

Answer: Authors thank the reviewer for the comment. After careful consideration we decided to keep the format of the diagrams, but to combine figures 3&4 and 5&6.  We believe in this way the graphs are representing the results in the best possible way.

  • Additionally, the numbers / proportions of cats that were negative for viral RNA in their blood, urine, and nasal swabs should be made clear

Answer: we think that the dots in the corresponding figures make it clear how many cats are negative.

  • Line 327 – please clarify that this is only 5 cats

Answer: done

  • Line 346 – given the superiority of virus neutralisation assays for serology please include the data here (in addition to reporting why neutralisations were not done for all serologic samples)

Answer: done. The reason was added in material and methods in chapter 2.1

  • Figure 9 – at what day was this cat sacrificed/ HP performed?

Answer: done. Histopathology was done and showed the same lesions as the other cats.

Discussion:

  • How do you reconcile the growth of FeMV in LLC-MK2 cells when presumably they don’t express catSLAM – this should be clarified?

Answer: as previously explained, morbilliviruses can grow in vitro on cells which do not express SLAM.

  • How do you reconcile the “weakly positive signal” in IFA in two cats at baseline. Although they are SPF cats, it is possible that they could have had prior exposure to FeMV, since it has not been an agent SPF cats are screened for historically. Why were they then not excluded from the infections

Answer: these weakly positive results came as a surprise. Since those cats were not viremic at day -11, we decided to keep them. Cats are a precious material difficult to obtain and not including them would have reduced the power of the experiment. The outcome of infection in those cats was similar to the others suggesting that they had not mounted a biologically relevant immune response against FeMV.

A comment was added in the discussion.

  • Line 407 – remove the period after “[29,35]”

Answer: done

  • Line 413-418 – can you expand upon this – do other morbilliviruses use SLAM from multiple species, could this just be adaptation to cell culture?

Answer: Yes, other morbilliviruses can use SLAMs from other species. We have a paragraph and a citation in the manuscript (lines 442-449), which describes this topic.

  • Line 446 – the viral loads in nasal swabs were very low – can you comment on this in the context of possible aerosol transmission

Answer: we agree with the reviewer’s comment. It would most likely be the main route of transmission but cannot be ruled out in the context of close proximity between cats. Comment added in the discussion.

  • A significant limitation is that the pathogenesis of the virus used to infect the cats (adapted to monkey cells) cannot be assumed to be the same as FeMV-GT 2, which in turn cannot be assumed to be the same as FeMV-GT1, which appears to be the more prevalence circulating strain based on existing data. WGS data must be provided for the virus used to perform the infections, and then the difference between this and wildtype strains must be clarified this as a limitation.

Answer: we agree with the reviewer’s comment. However, we would like to modulate it for the following reasons:

  • the passages on LLC-MK2 were the only method we could find to generate a stock with a titre and a volume sufficient to do the in vivo infection. It was a compulsory step at the time.
  • we don’t think that the number of passages was sufficient to completely attenuate a morbillivirus strain. Attenuation of morbilliviruses require usually a higher number of cell passages (Liu et al. 2016 Comp Immunol Microbiol).
  • the data we generated are very consistent with field observations of FeMV genotype 1 infections. This suggest that our strain was not fully attenuated. In addition, it suggests also that both genotypes may induce similar clinical signs and lesions. The next logical step is of course to run a similar study with FeMV-1.

We agree that it is a limitation and this will be added in the manuscript (end of discussion).

  • You mention the low level of SLAM sequence conservation between species (line 100), but then don’t discuss (just mention) the apparent use of dogSLAM by the virus. This requires clarification (eg. in the discussion lines 413-416).

Answer: The usage of SLAM from broader spectrum of carnivores is well known characteristics of morbilliviruses. Even CDV and MeV can broadly use non-host entry receptors from human and carnivores respectively. For further information, we included reference 29. in the manuscript.

Round 2

Reviewer 3 Report

Thank you for the opportunity to again review this manuscript. I will start by addressing some of my previous comments and your responses.

Firstly, I still struggle with justification (Lines 532-535) that what you saw in experimentally infected cats is consistent with what occurs in naturally developing infections – since no-one has documented natural infection from the time of infection and followed cats forward. Rather, the majority of naturally infected cats in the existing literature were likely chronically infected / shedding at the time that they were sampled, it is impossible to know if they had clinical signs of an acute illness when they were first infected. For example, lots of kittens have acute upper respiratory tract infections classically attributable to FHV-1 and FCV (and other classic feline URT pathogens), but it is possible that they had FeMV and no one realised. I would request that this part of the discussion is reworded. 

I would also request that the authors include a sentence in the discussion to clarify that sequencing the viral genome would be required to determine the genetic changes in the adaptation to cell culture from the wildtype virus. I appreciate the reference (Liu et al.) you have provided from other species in your response to my initial review – consider adding a sentence or two in to the discussion to clarify the numbers of passages required for attenuation as demonstrated with closely related morbilliviruses

Secondly, please expand on the discussion regarding the two weakly seropositive cats

Eg. for Line 500 –  you still have to consider that there are 2 possible explanations - false positive IFA results or true positive results ie. the cats had been exposed to the virus before and that this may have affected their response to experimental infections. Please add this additional level of detail to the discussion here 

Please also consider that SPF cats are not free of all microbes, they are just free of specific pathogens. Since FeMV is a newly discovered virus it is possible that SPF cat colonies could include cats that have / have been been exposed to FeMV. 

Thank you for providing more detail about primers, probes and the PCRs used in your response document. Please add all of this information to the methods section of the manuscript. 

Thank you for providing more details about the animal experiments. “Urine was collected by cystocentesis. Blood was collected from the jugular vein. Animals were sedated only if necessary, by using Domitor for sedation and Antisedan for awakening. Tissues were collected immediately after euthanasia.”

This too should be added to the methods section as it influences your results (eg. collection of urine by cystocentesis rather than bladder expression). 

Figures 4a, b and c are still too busy, displaying data in too many ways. Bar charts are designed to display categorical data – your data are continuous. Please remove the bar chart component, and rather just leave the dots (for individual cats) and the mean/median and SD lines – please also include what these lines represent in your Figure legend

Additional minor comments:

·      Line 188 – perhaps reword to “via an intravenous catheter”

·      Line 245 – do you mean “was not available”

·      Line 357 – please include the virus neutralization data

·      Line 503 – please add references to clarify that such lesions are not expected in SPF cats

·      Lines 542-543 – please add a reference if these data have been published elsewhere. If not already, will the data be published?

A few editorial things

·      Line 37 – add a period after the last reference for this sentence 

·      Line 55 – remove either the word “currently” or “yet” from this sentence, both are not needed

·      Line 57 – “6.7%” of what – perhaps clarify that that was the prevalence in that population

·      Line 59 – are you referring to “FeMV infection” here – please clarify

·      Line 109 – add the word “a” before ”a wide variety”

·      Table 1- please change “weighting” to “body weight” or “weighing”

·      Line 313 – please us a capital L for uL here

·      Figures 4a and b – please change ml to mL in the y axis label

·      Line 426 – please add a space after the period, and before the sentence starting “A possible ….”

·      Line 437 – please change “extend” to “extent"

Author Response

Answers to the second round of questions from reviewer No.3 are in the attached document.
